# Uncovering the Contribution of Moderate-Penetrance Susceptibility Genes to Breast Cancer by Whole-Exome Sequencing and Targeted Enrichment Sequencing of Candidate Genes in Women of European Ancestry

**DOI:** 10.3390/cancers14143363

**Published:** 2022-07-11

**Authors:** Martine Dumont, Nana Weber-Lassalle, Charles Joly-Beauparlant, Corinna Ernst, Arnaud Droit, Bing-Jian Feng, Stéphane Dubois, Annie-Claude Collin-Deschesnes, Penny Soucy, Maxime Vallée, Frédéric Fournier, Audrey Lemaçon, Muriel A. Adank, Jamie Allen, Janine Altmüller, Norbert Arnold, Margreet G. E. M. Ausems, Riccardo Berutti, Manjeet K. Bolla, Shelley Bull, Sara Carvalho, Sten Cornelissen, Michael R. Dufault, Alison M. Dunning, Christoph Engel, Andrea Gehrig, Willemina R. R. Geurts-Giele, Christian Gieger, Jessica Green, Karl Hackmann, Mohamed Helmy, Julia Hentschel, Frans B. L. Hogervorst, Antoinette Hollestelle, Maartje J. Hooning, Judit Horváth, M. Arfan Ikram, Silke Kaulfuß, Renske Keeman, Da Kuang, Craig Luccarini, Wolfgang Maier, John W. M. Martens, Dieter Niederacher, Peter Nürnberg, Claus-Eric Ott, Annette Peters, Paul D. P. Pharoah, Alfredo Ramirez, Juliane Ramser, Steffi Riedel-Heller, Gunnar Schmidt, Mitul Shah, Martin Scherer, Antje Stäbler, Tim M. Strom, Christian Sutter, Holger Thiele, Christi J. van Asperen, Lizet van der Kolk, Rob B. van der Luijt, Alexander E. Volk, Michael Wagner, Quinten Waisfisz, Qin Wang, Shan Wang-Gohrke, Bernhard H. F. Weber, Peter Devilee, Sean Tavtigian, Gary D. Bader, Alfons Meindl, David E. Goldgar, Irene L. Andrulis, Rita K. Schmutzler, Douglas F. Easton, Marjanka K. Schmidt, Eric Hahnen, Jacques Simard

**Affiliations:** 1Genomics Center, CHU de Québec-Université Laval Research Center, 2705 Laurier Boulevard, Quebec City, QC GIV 4G2, Canada; martine.dumont@crchudequebec.ulaval.ca (M.D.); charles.joly-beauparlant@crchudequebec.ulaval.ca (C.J.-B.); arnaud.droit@crchudequebec.ulaval.ca (A.D.); stephane.dubois@crchudequebec.ulaval.ca (S.D.); annie-c.c-deschesnes@crchudequebec.ulaval.ca (A.-C.C.-D.); penny.soucy@crchudequebec.ulaval.ca (P.S.); maxime.vallee@chu-lyon.fr (M.V.); frederic.fournier.4@ulaval.ca (F.F.); audrey@mims.ai (A.L.); 2Center for Familial Breast and Ovarian Cancer, Center for Integrated Oncology (CIO), Faculty of Medicine and University Hospital Cologne, University of Cologne, 50937 Cologne, Germany; nana.weber-lassalle@uk-koeln.de (N.W.-L.); corinna.ernst@uk-koeln.de (C.E.); rita.schmutzler@uk-koeln.de (R.K.S.); eric.hahnen@uk-koeln.de (E.H.); 3Department of Dermatology, University of Utah, Salt Lake City, UT 84103, USA; bingjian.feng@hsc.utah.edu (B.-J.F.); david.goldgar@hsc.utah.edu (D.E.G.); 4Huntsman Cancer Institute, University of Utah, Salt Lake City, UT 84112, USA; sean.tavtigian@hci.utah.edu; 5Family Cancer Clinic, The Netherlands Cancer Institute—Antoni van Leeuwenhoek Hospital, 1066 Amsterdam, The Netherlands; m.adank@nki.nl (M.A.A.); f.hogervorst@nki.nl (F.B.L.H.); l.vd.kolk@nki.nl (L.v.d.K.); 6Centre for Cancer Genetic Epidemiology, Department of Public Health and Primary Care, University of Cambridge, Cambridge CB1 8RN, UK; jma@ebi.ac.uk (J.A.); mkh39@medschl.cam.ac.uk (M.K.B.); sc2017@medschl.cam.ac.uk (S.C.); paul.pharoah@medschl.cam.ac.uk (P.D.P.P.); qw232@medschl.cam.ac.uk (Q.W.); dfe20@medschl.cam.ac.uk (D.F.E.); 7Cologne Center for Genomics (CCG), Faculty of Medicine, University Hospital Cologne, University of Cologne, 50931 Cologne, Germany; janine.altmueller@mdc-berlin.de (J.A.); holger.thiele@uni-koeln.de (H.T.); 8Institute of Clinical Molecular Biology, Department of Gynaecology and Obstetrics, University Hospital of Schleswig-Holstein, Campus Kiel, Christian-Albrechts University Kiel, 24105 Kiel, Germany; norbert.arnold@uksh.de; 9Division Laboratories, Pharmacy and Biomedical Genetics, Department of Genetics, University Medical Center Utrecht, 3584 Utrecht, The Netherlands; m.g.e.m.ausems@umcutrecht.nl; 10Institute of Human Genetics, Technische Universität München, 81675 Munich, Germany; riccardo.berutti@helmholtz-muenchen.de (R.B.); tim.strom@tum.de (T.M.S.); 11Lunenfeld-Tanenbaum Research Institute, Sinai Health System, Toronto, ON M5G 1X5, Canada; bull@lunenfeld.ca (S.B.); jessica.green@sinaihealth.ca (J.G.); gary.bader@utoronto.ca (G.D.B.); andrulis@lunenfeld.ca (I.L.A.); 12Dalla Lana School of Public Health, University of Toronto, Toronto, ON M5T 3M7, Canada; 13Division of Molecular Pathology, The Netherlands Cancer Institute—Antoni van Leeuwenhoek Hospital, 1066 Amsterdam, The Netherlands; s.cornelissen@nki.nl (S.C.); r.keeman@nki.nl (R.K.); mk.schmidt@nki.nl (M.K.S.); 14Precision Medicine and Computational Biology, Sanofi Genzyme, Cambridge, MA 02142, USA; michael.dufault@sanofi.com; 15Centre for Cancer Genetic Epidemiology, Department of Oncology, University of Cambridge, Cambridge CB1 8RN, UK; amd24@medschl.cam.ac.uk (A.M.D.); craig@srl.cam.ac.uk (C.L.); ms483@medschl.cam.ac.uk (M.S.); 16Institute for Medical Informatics, Statistics and Epidemiology, University of Leipzig, 04107 Leipzig, Germany; christoph.engel@imise.uni-leipzig.de; 17Centre of Familial Breast and Ovarian Cancer, Department of Medical Genetics, Institute of Human Genetics, University of Würzburg, 97074 Würzburg, Germany; gehrig@biozentrum.uni-wuerzburg.de; 18Department of Clinical Genetics, Erasmus University Medical Center, 3015 Rotterdam, The Netherlands; w.geurts-giele@erasmusmc.nl; 19Institute of Epidemiology, Helmholtz Zentrum München, German Research Center for Environmental Health, 85764 Neuherberg, Germany; christian.gieger@helmholtz-muenchen.de (C.G.); peters@helmholtz-muenchen.de (A.P.); 20Research Unit Molecular Epidemiology, Helmholtz Zentrum München, German Research Centre for Environmental Health, 85764 Neuherberg, Germany; 21Department of Molecular Genetics, University of Toronto, Toronto, ON M5S 1A8, Canada; kvn.kuang@mail.utoronto.ca; 22Institute for Clinical Genetics, Faculty of Medicine Carl Gustav Carus, Technische Universität Dresden, 01307 Dresden, Germany; karl.hackmann@uniklinikum-dresden.de; 23The Donnelly Centre, University of Toronto, Toronto, ON M5S 3E1, Canada; mohamed_helmy@bii.a-star.edu.sg; 24Bioinformatics Institute (BII), Agency for Science, Technology and Research (A*STAR), Singapore 138632, Singapore; 25Department of Computer Science, Lakehead University, Thunder Bay, ON P7B 5E1, Canada; 26Institute of Human Genetics, University Leipzig, 04103 Leipzig, Germany; julia.hentschel@medizin.uni-leipzig.de; 27Department of Medical Oncology, Erasmus MC Cancer Institute, 3015 Rotterdam, The Netherlands; a.hollestelle@erasmusmc.nl (A.H.); m.hooning@erasmusmc.nl (M.J.H.); j.martens@erasmusmc.nl (J.W.M.M.); 28Institute of Human Genetics, University of Münster, 48149 Münster, Germany; judit.horvath@ukmuenster.de; 29Department of Epidemiology, Erasmus MC University Medical Center, 3015 Rotterdam, The Netherlands; m.a.ikram@erasmusmc.nl; 30Institute of Human Genetics, University Medical Center Göttingen, 37075 Göttingen, Germany; silke.kaulfuss@med.uni-goettingen.de; 31German Center for Neurodegenerative Diseases (DZNE), Department of Neurodegenerative Diseases and Geriatric Psychiatry, Medical Faculty, University Hospital Bonn, 53127 Bonn, Germany; wolfgang.maier@ukb.uni-bonn.de; 32Department of Gynecology and Obstetrics, University Hospital Düsseldorf, Heinrich-Heine University Düsseldorf, 40225 Düsseldorf, Germany; niederac@med.uni-duesseldorf.de; 33Center for Molecular Medicine Cologne (CMMC), Cologne Center for Genomics (CCG), Faculty of Medicine, University Hospital Cologne, University of Cologne, 50931 Cologne, Germany; nuernberg@uni-koeln.de; 34Institute of Medical Genetics and Human Genetics, Charité-Universitätsmedizin Berlin, Corporate Member of Freie Universität Berlin, Humboldt-Universität zu Berlin and Berlin Institute of Health, 13353 Berlin, Germany; claus-eric.ott@charite.de; 35Department of Epidemiology, Institute for Medical Information Processing, Biometry and Epidemiology, Medical Faculty, Ludwig-Maximilians-Universität München, 80539 Munich, Germany; 36Division for Neurogenetics and Molecular Psychiatry, Medical Faculty, University of Cologne, 50937 Cologne, Germany; alfredo.ramirez@uk-koeln.de; 37Division of Gynaecology and Obstetrics, Klinikum Rechts der Isar der Technischen Universität München, 81675 Munich, Germany; juliane.ramser@mri.tum.de (J.R.); alfons.meindl@gmx.de (A.M.); 38Institute of Social Medicine, Occupational Health and Public Health, Faculty of Medicine, University of Leipzig, 04103 Leipzig, Germany; steffi.riedel-heller@medizin.uni-leipzig.de; 39Institute of Human Genetics, Hannover Medical School, 30625 Hannover, Germany; schmidt.gunnar@mh-hannover.de; 40Department of Primary Medical Care, University Medical Center Hamburg-Eppendorf, 20246 Hamburg, Germany; m.scherer@uke.de; 41Institute of Medical Genetics and Applied Genomics, University of Tübingen, 72076 Tübingen, Germany; antje.staebler@med.uni-tuebingen.de; 42Institute of Human Genetics, University Hospital Heidelberg, 69120 Heidelberg, Germany; christian.sutter@med.uni-heidelberg.de; 43Department of Clinical Genetics, Leiden University Medical Center, 2333 Leiden, The Netherlands; c.j.van_asperen@lumc.nl (C.J.v.A.); r.b.van_der_luijt@lumc.nl (R.B.v.d.L.); 44Department of Medical Genetics, University Medical Center, 3584 Utrecht, The Netherlands; 45Institute of Human Genetics, University Medical Center Hamburg-Eppendorf, 20246 Hamburg, Germany; a.volk@uke.de; 46Department of Neurodegenerative Diseases and Geriatric Psychiatry, University Hospital Bonn, 53127 Bonn, Germany; michael.wagner@ukbonn.de; 47Department of Human Genetics, Amsterdam UMC, Vrije Universiteit Amsterdam, 1081 Amsterdam, The Netherlands; q.waisfisz@vumc.nl; 48Department of Gynaecology and Obstetrics, University of Ulm, 89081 Ulm, Germany; shan.wang-gohrke@uniklinik-ulm.de; 49Institute of Human Genetics, Regensburg University, 93053 Regensburg, Germany; bweb@klinik.uni-regensburg.de; 50Institute of Clinical Human Genetics, University Hospital Regensburg, 93053 Regensburg, Germany; 51Department of Pathology, Department of Human Genetics, Leiden University Medical Center, 2333 Leiden, The Netherlands; p.devilee@lumc.nl; 52Department of Oncological Sciences, University of Utah School of Medicine, Salt Lake City, UT 84132, USA; 53Department of Computer Science, University of Toronto, Toronto, ON M5S 3E1, Canada; 54Princess Margaret Research Institute, University Health Network, Toronto, ON M5G 0A3, Canada; 55Division of Psychosocial Research and Epidemiology, The Netherlands Cancer Institute—Antoni van Leeuwenhoek Hospital, 1066 Amsterdam, The Netherlands; 56Department of Molecular Medicine, Faculty of Medicine, Université Laval, Quebec, QC G1V 0A6, Canada

**Keywords:** breast cancer, genetic susceptibility, whole-exome sequencing, moderate-penetrance genes

## Abstract

**Simple Summary:**

Genetic variants explaining approximately 40% of familial breast cancer risk have been identified, thus leaving a significant fraction of the heritability of this disease still unexplained. The exact nature of this missing fraction is unknown; more extensive sequencing efforts could potentially identify new moderate-penetrance breast cancer risk alleles. The aim of this study was to perform a large-scale whole-exome sequencing study, followed by a targeted validation, in breast cancer patients and healthy women of European descent. We identified 20 novel genes with modest evidence of association (*p*-value < 0.05) for either overall or subtype-specific breast cancer; however, much larger studies are needed to confirm the exact role of these genes in susceptibility to breast cancer.

**Abstract:**

Rare variants in at least 10 genes, including *BRCA1*, *BRCA2, PALB2, ATM,* and *CHEK2*, are associated with increased risk of breast cancer; however, these variants, in combination with common variants identified through genome-wide association studies, explain only a fraction of the familial aggregation of the disease. To identify further susceptibility genes, we performed a two-stage whole-exome sequencing study. In the discovery stage, samples from 1528 breast cancer cases enriched for breast cancer susceptibility and 3733 geographically matched unaffected controls were sequenced. Using five different filtering and gene prioritization strategies, 198 genes were selected for further validation. These genes, and a panel of 32 known or suspected breast cancer susceptibility genes, were assessed in a validation set of 6211 cases and 6019 controls for their association with risk of breast cancer overall, and by estrogen receptor (ER) disease subtypes, using gene burden tests applied to loss-of-function and rare missense variants. Twenty genes showed nominal evidence of association (*p*-value < 0.05) with either overall or subtype-specific breast cancer. Our study had the statistical power to detect susceptibility genes with effect sizes similar to *ATM, CHEK2*, and *PALB2*, however, it was underpowered to identify genes in which susceptibility variants are rarer or confer smaller effect sizes. Larger sample sizes would be required in order to identify such genes.

## 1. Introduction

Breast cancer is the most common cause of cancer-related death and the most frequently diagnosed cancer among women worldwide [1]. Currently, approximately 40% of familial breast cancer risk is explained by a combination of common low-penetrance variants [2,3], together with coding rarer variants in predisposition genes, such as *BRCA1*, *BRCA2, PALB2, ATM*, and *CHEK2* that confer higher risks [4]. Thus, a significant fraction of the heritability of this disease is still unexplained. A better understanding of the genetic risk factors contributing to breast cancer can improve risk prediction, and hence better inform screening and prevention strategies, and may also inform understanding of the biology underlying breast cancer predisposition.

While linkage studies have identified high-risk breast cancer susceptibility genes, and genome-wide association studies (GWAS) using single-nucleotide polymorphism (SNP) arrays have identified many common variants conferring modest risks, neither of these approaches are powerful enough to identify rarer variants conferring moderate risk. Several genes associated with moderate risk (two- to four fold) of breast cancer have been identified by candidate gene sequencing approaches. These targeted approaches are limited to our understanding of the pathways involved in breast cancer etiology, and have thus mainly identified new breast cancer susceptibility alleles within genes either coding for proteins that interact with BRCA1 or BRCA2 (e.g., *PALB2*), or other genes involved in DNA repair processes, such as *ATM* and *CHEK2*. Since most genes have not yet been subjected to large-scale sequencing studies, it is possible that other susceptibility genes exist, which would explain some of the residual heritability of breast cancer.

Recently, whole-exome sequencing (WES) has provided a comprehensive, agnostic approach to exploring associations between rarer coding variants and disease. Compared to whole-genome sequencing (WGS), WES has a substantially lower cost and generates results that are more easily interpretable [5]. In the last few years, a number of WES studies have been carried out for breast cancer [6,7]. These studies rely on the sequencing of cases enriched for family history of disease, and matched controls, sometimes in combination with analysis of the segregation of the variants associated with disease in families. However, very few novel susceptibility genes have been robustly identified by these studies. This lack of success may, in part, be due to the lack of statistical power: since deleterious variants in most genes are likely to be very rare, studies need to be very large to detect them. More recently, the largest WES study for overall breast cancer reported increased breast cancer risk associated with *ATM*, *CHEK2*, *PALB2*, and *MSH6* [8]. However, while the first three genes are already well established, the association with *MSH6* has not been definitively replicated in large, targeted sequencing studies. 

In an attempt to overcome these shortcomings, we designed a large-scale two-stage WES study. The discovery step was carried out using WES data from 1528 breast cancer cases enriched for genetic susceptibility to the disease, based on early onset breast cancer, a family history of disease, or bilateral breast cancer, and 3733 geographically matched, unaffected controls. The validation step assessed the associations between the presence of loss-of-function (LoF) and rare missense variants in 198 selected candidate genes and the risk of breast cancer overall, as well as estrogen receptor (ER) disease subtypes, in a replication set of 6211 breast cancer cases and 6019 controls. In addition to these genes, a panel of 32 known or suspected breast cancer susceptibility genes was also included in this targeted enrichment sequencing study.

## 2. Materials and Methods

### 2.1. Discovery Stage—Whole-Exome Sequencing (WES) Analysis

#### 2.1.1. Studies and Datasets

For the discovery stage, WES was performed on 1528 breast cancer cases, and the resulting data were compared with whole-exome data obtained from 3733 unaffected controls (3483 unaffected women from four studies, as described below, and 250 unaffected men from GoNL trios) (Appendix A). Briefly, breast cancer cases were selected from two clinic-based studies: The German Consortium for Hereditary Breast and Ovarian Cancer (GC-HBOC) (1021 cases) [9], with half of the samples originating from the Cologne region (490 samples) and the other half from the Munich area (531 samples), and The Dutch Familial Bilateral Breast Cancer Study (DFBBCS) (511 cases) [10,11,12]. All subjects had previously tested negative for the presence of *BRCA1* and *BRCA2* germline mutations in a clinical genetic setting. All subjects from the GC-HBOC had a family history of breast cancer, while all subjects from the DFBBCS had bilateral breast cancer. Control whole-exome data were obtained from four studies: The Rotterdam Study [13], The Genome of the Netherlands Project (GoNL) [14,15], The German Study on Ageing, Cognition, and Dementia (AgeCoDe) [16], and The KORA Study [17]. A more detailed description of these studies and datasets is included in Appendix A. All study subjects were recruited based on protocols approved by the Institutional Review Boards at each participating institution, and all subjects provided written informed consent. 

#### 2.1.2. Selection of Breast Cancer Cases 

Breast cancer cases were selected according to the following criteria: early onset (<50 years of age) and a family history of one or more first-degree relatives with breast cancer diagnosed before 50 years of age, or two relatives diagnosed under the age of 60. A scoring system was established for the selection of families based on the assumption that rare, moderate-risk variants might be expected to confer a higher relative risk at young ages, as is the case for most known susceptibility genes [10]. In addition, the number of affected relatives weighted by their degree of relationship to the case was taken into account. Briefly, families were scored according to the following: (a) For each case in the family, a score of 1 was given if the case was diagnosed at age 50 or older, a score of 1.5 was given if diagnosed between 40–49 years, and a score of 2 if diagnosed at <40 years of age. (b) Cases were then weighted by their degree of relationship to the index case, where a weight of 1 was attributed for the index case, 0.5 for first-degree relatives, and 0.25 for second-degree relatives. (c) For bilateral cases, both cancers were included (thus giving a score of 2). (d) The total score for the family was obtained by the sum of these scores. Carriers of deleterious germline mutations in *BRCA1* and *BRCA2*, and *CHEK2* c.1100delC carriers were excluded. Related samples were identified on the basis of genotype data, and the individual with the youngest age of onset was chosen. The score distribution among breast cancer cases was as follows: 0.6% with a score below 2; 16.7% with a score between 2 and <2.5; 33.6% with a score between 2.5 and <3; 25.5% with a score between 3 and <3.5; 14.1% with a score between 3.5 and <4; 9.5% with a score ≥4.

#### 2.1.3. Library Preparation, High-Throughput Sequencing, and Bioinformatics Analysis

Libraries for samples from GC-HBOC cases were prepared using the Agilent SureSelectXT Human All Exon V5 capture kit, while libraries for samples from DFBBCS cases were prepared using NimbleGen SeqCap EZ Exome Library v3.0. (Appendix A). Barcoded libraries were sequenced on an Illumina HiSeq2500 for paired-end 100 bp sequencing. Available sequencing data from controls was obtained from other studies. These data were generated using the following capture technologies: NimbleGen SeqCap EZ Exome V2, NimbleGen SeqCap EZ Exome V3, Agilent SureSelect Human All Exon V3 and V5 (Appendix A). Whole-exome data from cases and controls were demultiplexed and aligned to the reference genome using BWA-MEM to create aligned BAM files, followed by base quality score recalibration. The analysis of sequencing data from cases and controls was restricted to the region of 34.5 Mb common to the different capture technologies used. For breast cancer cases, a mean coverage of 100× was obtained with >97.5% of bases above 10×. Variant calling was performed using a pipeline based on the GATK HaplotypeCaller. Details of library preparation and bioinformatics analysis including variant calling and quality control procedures are given in Appendix A.

#### 2.1.4. Variant Filtering and Gene Prioritization

Five research groups involved in this project (CHU de Québec-Université Laval Research Centre, Huntsman Cancer Institute, University of Utah, Munich Technical University, University of Toronto, University of Cambridge) performed variant filtering and gene prioritization using different approaches according to their expertise with regard to such analyses (Appendix A). Variant filtering and gene prioritization strategies are summarized in Figure 1 and in Appendix A. Most variant filtering strategies are similar for all groups, reflecting current best practices. 

#### 2.1.5. Aggregation of Gene Lists

Following these gene prioritization analyses, the results from each group underwent manual scrutiny and were then merged into a final list of candidate genes for validation. The selection process first included the top 30-ranked genes from each group. Each group then nominated additional putative breast cancer susceptibility genes according to their analysis and expertise. Finally, the list was augmented by the inclusion of genes that were among the top 500-ranked by at least four groups. Through this selection process, 198 candidate genes identified in the WES were selected for targeted resequencing (Appendix A). In addition to these candidate genes, this targeted sequencing enrichment stage also included 32 genes for which there was prior evidence of association with breast cancer risk [4,18]. These 32 genes were selected based on their *a priori* associations with the disease, and were not considered in the individual gene ranking process.

### 2.2. Validation Stage—Targeted Enrichment Sequencing

#### 2.2.1. Sample Sets

A total of 12,655 samples, including 6518 women affected with breast cancer, and 6137 controls, were selected for the validation stage. These were drawn from five sources, namely: GC-HBOC [9] (n = 5966, 3199 cases and 2767 controls); a Dutch validation set comprising samples from two studies, the Amsterdam Breast Cancer Study-Familial (ABCS-F) [10] and the Rotterdam Breast Cancer Study (RBCS) [19] (n = 1941, 979 cases and 962 controls); Studies of Epidemiology and Risk factors in Cancer Heredity (SEARCH) [20,21,22] (n = 2637, 1289 cases and 1348 controls); the Ontario Familial Breast Cancer Registry (OFBCR) [23] (n = 1191, 600 cases and 591 controls) and CARTaGENE (CaG) [24] (n = 920, 451 cases and 469 controls). The GC-HBOC samples did not overlap with those used for the discovery step. A more detailed description of these studies is included in Appendix A. All study subjects were recruited based on protocols approved by the Institutional Review Boards at each participating institution, and all subjects provided written informed consent. 

#### 2.2.2. Power Calculations

To perform power calculations for the replication stage, we assumed a significance level of *p*-value < 10^−4^. The standard “exome-wide” significance level is *p* < 2.5 × 10^−6^; this is equivalent to assuming a prior probability 40-fold higher than an average gene. This is also the level proposed by Easton et al. [4] for assessing the evidence for candidate DNA repair genes included in gene panels. To allow for the oversampling of cases with a family history, bilaterality, or early age at onset, we assumed that the effect size (log odds ratio) was increased by a factor of 1.5. Based on these assumptions, the power exceeded 50% for genes with a carrier frequency >0.005 and an odds ratio (OR) in the population >2, or a carrier frequency >0.001 and an OR > 4. Based on the effect sizes and European allele frequencies [18], the power of our study to detect genes with the same characteristics as *ATM*, *CHEK2*, and *PALB2* was 0.44, 1.0 and 0.95, respectively. However, the power was low to detect genes with the characteristics of *RAD51C* (0.0039), *RAD51D* (0.0025), or *BARD1* (0.017).

#### 2.2.3. Library Preparation and High-Throughput Sequencing

Library preparation was performed at the Cologne Center for Genomics, the Center for Familial Breast and Ovarian Cancer, and the Genomics Centre at CHU de Québec-Université Laval Research Center. Sequencing was performed at two centers: the Cologne Center for Genomics, Cologne, Germany, using Illumina HiSeq 4000 sequencers for paired-end 75 bp sequencing (mean coverage 140×); and the Genomics Centre at CHU de Québec-Université Laval Research Center, Quebec City, Canada, using Illumina HiSeq 2500 for paired-end 100 bp sequencing (mean coverage 185×). Additional details on library preparation and quality control procedures are given in the Appendix A. 

#### 2.2.4. Bioinformatics and Statistical Analyses

Bioinformatics analyses were performed on the Compute Canada supercomputer, Graham. A pipeline was set up following GATK Best Practices [25]. GATK Unified Genotyper was used for variant calling, and variants were annotated and quality-controlled using VICTOR (variant interpretation for clinical testing or research), and both gene-level and variant-level significances were computed using PERCH (polymorphism evaluation, ranking and classification for heritable traits) [26]. PERCH quantitatively integrated deleteriousness prediction, allele frequency information, rare variant association analysis, biological relevance assessment, and the quality of variant calls. Additional details on the bioinformatics pipeline and variant calling procedure are given in the Appendix A. Following these quality control and filtering steps, data from a total of 12,230 samples were further analyzed, including 6211 breast cancer cases and 6019 controls. 

The principal association analysis was a gene burden analysis, in which the genotypes were collapsed into a 0/1 variable for each gene, according to whether or not a deleterious variant was carried. For each gene, variant carriers were defined using the VICTOR software suite. LoF variants were defined as (1) StopGain or FrameShift variants, excluding those affecting only the last 5% of coding sequences (CDS); (2) SpliceSite variants that include coding and non-coding regions; and (3) variants that inhibit transcription or translation. Rare missense variants included in these analyses were those showing a deleteriousness score according to the BayesDel algorithm [26] (Appendix A). A final list of 7437 variants was established, including LoF variants (n = 1696) and rare missense variants (n = 5741). Among these, 25 LoF variants had a minor allele frequency (MAF) between 0.01 and 0.001, and 98.5% of LoF variants had an MAF < 0.001; in contrast, 21 missense variants had an MAF between 0.01 and 0.001, and the remaining 99.6% of missense variants had an MAF < 0.001.

ORs and 95% confidence intervals (CI) associated with carrier status were computed using logistic regression, adjusting for study population as a covariate, and *p*-values were derived from Wald tests. For analyses performed by individual study, *p*-values were computed via Fisher’s exact tests. All statistical analyses were performed in R v3.6.0.

## 3. Results

A final list of 7437 variants including LoF (n = 1696) and rare missense variants (n = 5741) were included in the gene burden analyses. Quantile–quantile (Q–Q) plots are shown in Appendix A for LoF variants, and Appendix A for missense variants. When known breast cancer susceptibility genes were excluded, there was no evidence of inflation in the test statistics.

### 3.1. Known or Suspected Breast Cancer Susceptibility Genes

Targeted enrichment sequencing of the panel of 32 known or suspected breast cancer susceptibility genes confirmed the associations of known breast cancer susceptibility genes *ATM, CHEK2*, and *PALB2* (Figure 2, Table 1, Appendix A). It should be noted that these analyses did not include *BRCA1* or *BRCA2*, as our original study design excluded samples from families that carried mutations in either of these genes.

#### 3.1.1. Loss-of-Function Variants

Significant associations for LoF variants with overall breast cancer risk were observed for *CHEK2* (OR = 3.47 (CI 95% 2.33–5.15), *p*-value = 7.62 × 10^−10^); *ATM* (OR = 3.55 (CI 95% 2.16–5.82), *p*-value = 5.44 × 10^−7^); and *PALB2* (OR = 3.95 (CI 95% 1.98–7.88), *p*-value = 9.68 × 10^−5^), while more modest evidence of overall risk association was observed for *TP53* (OR = 10.16 (CI 95% 1.31–78.67, *p*-value = 0.026) (Figure 2 and Table 1). No nominally significant associations were observed for the remaining 28 genes in this panel. Of the four genes that were associated with breast cancer overall, *CHEK2* and *PALB2* also showed evidence of association with both ER-positive and ER-negative disease (*CHEK2* ER-positive: OR = 4.44 (CI 95% 2.82–6.99), *p*-value = 1.23 × 10^−10^ and ER-negative: OR = 2.65 (CI 95% 1.24–5.70), *p*-value = 0.012; *PALB2* ER-positive: OR = 5.27 (CI 95% 2.39–11.61), *p*-value = 3.83 × 10^−5^ and ER-negative: OR = 6.45 (CI 95% 2.31–17.98), *p*-value = 3.72 × 10^−4^) while *ATM* showed evidence of an association with ER-positive disease (OR = 4.06 (CI 95% 2.37–6.97), *p*-value = 3.39 × 10^−7^, *p*-diff = 0.026 for difference with ER-negative disease) (Figure 2, Table 1). Among the genes that had no evidence of association with overall breast cancer risk, *RAD51D* showed some evidence of association with ER-negative disease (OR = 11.26 (CI 95% 2.60–48.68), *p*-value = 1.19 × 10^−3^, *p*-diff = 0.049 for difference with ER-positive disease) (Figure 2, Table 1). LoF variants in *MSH6* showed evidence of a negative association with breast cancer risk. This effect was observed for overall breast cancer risk (OR = 0.36 (CI 95% 0.21–0.62), *p*-value = 2.07 × 10^−4^) as well as for ER-positive specific disease (OR = 0.42 (CI 95% 0.22–0.81), *p*-value = 9 × 10^−3^) (Table 1).

#### 3.1.2. Rare Missense Variants

As described in the Materials and Methods section, rare missense variants included in these analyses were those showing a deleteriousness score according to the BayesDel algorithm [26] (Appendix A). In the panel of 32 known or suspected breast cancer susceptibility genes, there was evidence of an association with overall breast cancer risk for rare missense variants in four genes: *ATM* (OR = 1.67 (CI 95% 1.32–2.10), *p*-value = 1.56 × 10^−5^); *CHEK2* (OR = 1.95 (CI 95% 1.35–2.82), *p*-value = 3.43 × 10^−4^); *TP53* (OR = 2.22 (CI 95% 1.26–3.93), *p*-value = 5.86 × 10^−3^) and *MEN1* (OR = 5.93 (CI 95% 1.33–26.50), *p*-value = 0.020) (Figure 2 and Appendix A). The results of the tumor ER-subtype analyses are shown in Appendix A. Among genes that were not associated with overall breast cancer risk, missense variants in *BARD1* showed some evidence of association with ER-positive disease (OR = 2.14 (CI 95% 1.13–4.04), *p*-value = 0.020), while an association with ER-negative breast cancer was observed for missense variants in *BRIP1* (OR = 3.78 (CI 95% 1.23–11.55), *p*-value = 0.020) (Figure 2, Appendix A).

### 3.2. Validation of Candidate Genes Identified at the Discovery Stage

Analysis of the 198 genes selected from the WES analysis identified 20 potential candidate susceptibility genes with evidence of association with breast cancer (*p*-value < 0.05) (Figure 3, Table 2 and Appendix A). 

#### 3.2.1. Loss-of-Function Variants

LoF variants in three genes (*ZFAND1*, *TYRO3*, and *TMEM206/PACC1*) displayed modest evidence of an association with breast cancer risk overall (*p*-value < 0.05) (Figure 3 and Table 2), with effect sizes for *ZFAND1* of 1.73 (CI 95% 1.12–2.68, *p*-value = 0.014), *TYRO3* of 1.40 (CI 95% 1.06–1.83, *p*-value = 0.016), and *TMEM206/PACC1* of 1.70 (CI 95% 1.11–2.62, *p*-value = 0.016). Of these genes, two were also associated with ER-negative breast cancer risk: *ZFAND1* (OR = 2.96 (CI 95% 1.59–5.50), *p*-value = 6.37 × 10^−4^) and *TYRO3* (OR = 1.66 (CI 95% 1.03–2.68), *p*-value = 0.038). A negative overall association with breast cancer risk was observed for *EML5* (OR = 0.41 (CI 95% 0.18–0.94), *p*-value = 0.035). Tumor subtype specific analyses also revealed that, among the genes that showed no evidence of an association with overall breast cancer risk, two genes showed some evidence of an association with ER-negative breast cancer, namely *DNAH11* (OR = 3.00 (CI 95% 1.42–6.32), *p*-value = 3.88 × 10^−3^, *p*-value for difference with ER-positive disease is 0.012) and *PARP2* (OR = 6.89 (CI 95% 1.11–42.78), *p*-value = 0.038), while *LAMC3* (OR = 3.14 (CI 95% 1.20–8.20), *p*-value = 0.020), *MTMR11* (OR = 1.49 (CI 95% 1.03–2.17), *p*-value = 0.037), *EPN3* (OR = 1.97 (CI 95% 1.03–3.76), *p*-value = 0.039), and *SLC22A10* (OR = 3.70 (CI 95% 1.01–13.57), *p*-value = 0.048) showed evidence of an association with ER-positive disease (Figure 3, Table 2).

#### 3.2.2. Rare Missense Variants

There was modest evidence of an association with overall breast cancer risk for rare missense variants (Table 2) in three genes: *TMEM161A* (OR = 2.56 (CI 95% 1.19–5.51), *p*-value = 0.016), *SIPA1L1* (OR = 1.68 (CI 95% 1.06–2.67), *p*-value = 0.026), and *ERCC2* (OR = 1.45 (CI 95% 1.01–2.08), *p*-value = 0.047). Missense variants in *TYRO3* showed evidence of a negative association with risk (OR = 0.29 (CI 95% 0.10–0.89), *p*-value = 0.031), in contrast with the effects observed for LoF variants. Eleven missense variants were observed for this gene, each only once, either in cases or in controls; the exception was variant p.R750C rs145529129, which was predominantly observed in unaffected controls, and thus, appears to explain the majority of the observed protective association. Of the three genes associated with overall breast cancer risk, *TMEM161A* was also associated with ER-positive breast cancer risk (OR = 2.73 (CI 95% 1.11–6.71), *p*-value = 0.029). ER-subtype-specific analyses further revealed associations with ER-negative disease for *SLC22A10* (OR = 20.76 (CI 95% 2.03–211.95), *p*-value = 0.011), *PHAX* (OR = 24.27 (CI 95% 2.42–243.37), *p*-value = 6.69 × 10^−3^), *SMARCA2* (OR = 4.92 (CI 95% 1.47–16.48), *p*-value = 9.69 × 10^−3^), *EML5* (OR = 2.74 (CI 95% 1.25–6.01), *p*-value = 0.012), *NTRK3* (OR = 3.17 (CI 95% 1.11–9.11), *p*-value = 0.032), and *MED23* (OR = 2.29 (CI 95% 1.01–5.18), *p*-value = 0.048), while *RNF175* (OR = 2.59 (CI 95% 1.12–6.00), *p*-value = 0.026) and *NCKAP1L* (OR = 2.57 (CI 95% 1.12–5.91), *p*-value = 0.027) were associated with ER-positive breast cancer. As mentioned in the previous section, we observed modest evidence of a negative association with overall breast cancer risk for LoF variants in *EML5*, which is in contrast with the observed association of missense variants in this gene with ER-negative disease.

#### 3.2.3. Combined Analysis of LoF and Rare Missense Variants

The combined analysis of LoF and rare missense variants (Table 2) identified a new association for *ABCC2* (*p*-value = 0.038) with overall breast cancer, and also identified additional distinct associations for two genes for which evidence of associations had been observed in either separate LoF or missense variants analyses, namely, a modest association of *SMARCA2* with overall breast cancer (OR = 2.43 (CI 95% 1.01–5.82), *p*-value = 0.046), and one for *TMEM206/PACC1* with ER-negative breast cancer (OR = 2.26 (CI 95% 1.13–4.51), *p*-value = 0.021). Moreover, the combined analysis also identified associations (*p* < 0.05), which had also been observed in separate LoF or missense variants analyses, for the following genes: *ZFAND1* (*p*-value = 0.014), *TMEM206/PACC1* (*p*-value = 0.011), *TMEM161A* (*p*-value = 0.014), and *SIPA1L1* (*p*-value = 0.026) with overall breast cancer; *ZFAND1* (*p*-value = 1.78 × 10^−3^), *DNAH11* (*p*-value = 0.037), *SLC22A10* (*p*-value = 0.011), *PHAX* (*p*-value = 5.50 × 10^−3^), *SMARCA2* (*p*-value = 3.40 × 10^−3^), and *NTRK3* (*p*-value = 0.032) with ER-negative breast cancer subtype; and *MTMR11* (*p*-value = 0.032), *TMEM161A* (*p*-value = 0.015), *RNF175* (*p*-value = 0.031), and *NCKAP1L* (*p*-value = 0.048) for ER-positive breast cancer subtype.

#### 3.2.4. Individual LoF and Missense Variant Analysis

As mentioned in the Materials and Methods section, the main association analysis performed in our study was a gene burden analysis. This approach increases the power to detect an association, and has been used by a vast majority of association studies, including those recently published by Dorling et al. [18] and Hu et al. [27]. Nevertheless, it has been shown that single variants can show evidence of association when a weaker signal is obtained at the gene level [28,29]. We therefore examined whether certain individual LoF or missense variants with higher frequencies showed evidence of association, and whether they were driving the observed associations for a given gene. In this analysis, we only examined individual variants observed in at least three carriers, at least two of which were observed in breast cancer cases. As shown in Appendix A, a total of seven LoF variants in six different genes showed evidence of association with overall breast cancer (nominal *p* < 0.05). Three of these variants were observed in known breast cancer susceptibility genes, one in *ATM* (c.1564_1565delGA, OR = 8.77 (CI 95% 1.11–69.29), *p*-value = 0.040), which is reported as pathogenic in the ClinVar database [30], and two in *CHEK2* (c.1100delC, OR = 3.08 (CI 95% 2.01–4.3), *p*-value = 2.83 × 10^−7^; c.444 + 1G > A, OR = 8.12 (CI 95% 1.03–64.13), *p*-value = 0.047). Three other LoF variants showing evidence of association in our single variant analysis were located in the top three candidate genes identified through the gene burden test: *ZFAND1* (c.139-3_139delCAGG, OR = 1.76 (CI 95% 1.11–2.81), *p*-value = 0.017), *TMEM206/PACC1* (c.631delC, OR = 1.81 (CI 95% 1.15–2.86), *p*-value = 0.011), and *TYRO3* (c.308_308 + 1insCCTGAAGTCA, OR = 1.55 (CI 95% 1.15–2.09), *p*-value = 3.93 × 10^−3^). These three variants have a much higher frequency than other such variants identified in these genes. They clearly contribute to the association observed at the gene level. Lastly, one LoF variant in the *TRPM4* gene had a very high OR, while no evidence of association was observed for this gene when all LoF variants were analyzed in aggregate (*TRPM4*, c.2254C > T, OR = 10.64 (CI 95% 1.37–82.46), *p*-value = 0.024). A similar analysis performed on rare missense variants (Appendix A) identified only three variants with nominally significant *p*-values: one in *ATM* (p.L2307F, OR = 5.94 (CI 95% 2.06–17.16), *p*-value = 9.85 × 10^−4^), one in *PMS2* (p.I18T, OR = 4.74 (CI 95% 1.02–22.01), *p*-value = 0.047), and one in *RIC1* (p.R635H, OR = 6.07 (CI 95% 1.36–27.17), *p*-value = 0.018). With the exception of the *ATM* gene, *PMS2* and *RIC1* showed no evidence of association in the gene burden test. Even though these variants showed a deleteriousness score according to the BayesDel algorithm, caution should be taken when interpreting missense variants because their impact may differ greatly depending on their position in the gene (e.g., conserved or functional domains).

## 4. Discussion

Our study identified 20 new genes showing modest evidence of an association either with overall disease and/or with ER-positive breast cancer or ER-negative breast cancer (*p*-value < 0.05). These results are based on the analysis of LoF variants and rare missense variants. Of these twenty, three showed evidence of association with LoF variants for overall breast cancer (*ZFAND1, TYRO3*, *TMEM206/PACC1*), and a further six with breast cancer subtypes (*DNAH11, PARP2, LAMC3, MTMR11, EPN3, SLC22A10*). When missense variants were assessed, three further associations were observed for overall breast cancer (*TMEM161A*, *SIPA1L1*, *ERCC2)*, and a further seven were observed for subtypes (*RNF175, NCKAP1L, PHAX, SMARCA2, EML5*, *NTRK3, MED23)*. One additional association was observed when missense and LoF variants were combined (*ABCC2*). Detailed information on these 20 genes, including a description of the known functions and pathways in which they are involved, as well as the number of LoF and missense variants observed in the different analyses, are provided in Appendix A. It should be noted that none of the genes reached the *p* < 10^−4^ threshold in the replication stage, suggested for candidate genes [4]. Moreover, the number of associations observed at the *p* < 0.05 level for each analysis is actually less than the number that would be expected by chance (10, given 198 genes). Thus, larger replication studies will be required to confirm or refute these associations.

Among these genes, *ZFAND1, TYRO3*, and *TMEM206/PACC1* showed evidence of association with LoF variants for overall breast cancer. In *ZFAND1*, one variant (c.139-3_139delCAGG, rs1260212806) explained the majority of the observed association. It is located in a splice site junction (acceptor site). This variant was observed only in the German study, so it could be more frequent in this population. The *TMEM206/PACC1* gene is involved in the progression of colorectal cancer by accelerating cell proliferation and promoting cell migration and invasion [31]. Five LoF variants were identified in this gene, one of which was observed in 93% of the carriers (c.631delC, rs532279691). *TYRO3* has been found to be upregulated in various cancers, including AML, CML, multiple myeloma, melanoma, as well as uterine endometrial cancers [32,33,34]. Eight LoF variants were identified in *TYRO3*, including one recurrent variant c.308_308 + 1insCCTGAAGTCA (rs773930671), accounting for 85% of carriers of LoF variants in this gene. Our analysis of missense variants in *TYRO3* revealed evidence of a protective effect, which is opposite to that observed for LoF variants. Eleven rare missense variants were identified in our study, the majority only being observed once, and seven of them only observed in controls. 

Tumor subtype-specific analyses were performed and, as described below, some evidence of ER-specific associations was observed. It should be noted that ER status was only available for 3572 (57.51%) breast cancer cases (808 ER-negative cases and 2764 ER-positive cases). Tumor subtype analyses of LoF variants revealed evidence of association for *ZFAND1, TYRO3, DNAH11*, and *PARP2* with ER-negative breast cancer, with *ZFAND1* showing the strongest association (OR = 2.96 (CI 95% 1.59–5.50), *p*-value = 6.37 × 10^−4^). The observed association for this gene is mainly due to the variant rs1260212806, previously mentioned for breast cancer overall. A few reports have linked variants (other than those observed in the current study) in *DNAH11* with breast cancer risk, but these reports were based on comparatively small sample sizes (c.2081_2082del (p.Val694Glyfs*2)) [35] and studies involving specific populations (G > A (rs2494938-India)) [36]. More recently, large-scale GWAS studies performed by our group through the Breast Cancer Association Consortium (BCAC) identified a common variant located at the 3′-UTR region of this gene (rs7971) to be statistically significantly associated with breast cancer risk (*p*-value = 1.9 × 10^−8^) [3]. Although further validation is needed, these observations seem to indicate that this locus may comprise both common and rare variants, with some evidence of association with breast cancer risk.

On the other hand, evidence of association with ER-positive disease with regard to LoF variants was observed for *MTMR11, LAMC3*, and *EPN3*. For *MTMR11,* the observed association is mainly due to one recurrent variant, c.14_15insG (rs587606143), which accounts for 94% of carriers of a LoF variant in this gene. Interestingly, a locus at 1q12-q21, which includes *MTMR11*, was identified by GWAS as being associated with mammographic density (rs11205303, OR = 0.73 (95% CI = 0.66–0.80), *p* = 2.64 × 10^−11^) [37]. Large-scale GWAS and fine-mapping studies performed in the BCAC have shown that common variants at 1q21.2 are significantly associated with overall and ER-positive breast cancer risk (overall breast cancer *p*-value = 9 × 10^−14^; ER + breast cancer *p*-value = 1 × 10^−12^) [3,38]. One study has also reported altered expression levels of *MTMR11* in breast cancer cells [39]. These observations make this locus an interesting candidate for further validation and characterization with regard to breast cancer. Although a modest association was observed in the current study for *SLC22A10,* the frequency of rare variants in cases and controls was very low for this gene (Appendix A).

In addition to clinical estrogen receptor subtyping, analyses performed on the molecular subtypes of breast cancer, namely, luminal A, luminal B, HER2-enriched, basal-like, and normal-like, would have been relevant in the context of this analysis, given the recognized heterogeneity of ER-positive and ER-negative tumors. However, the limited number of samples for which we had data did not allow us to perform a meaningful analysis of these molecular subtypes.

A number of genes showed an excess of missense variants (MAF < 0.01) classified as deleterious using bioinformatics in silico methods, without showing a corresponding excess of LoF variants. The best putative candidates in this group were *TMEM161A*, *ERCC2*, and *SIPA1L1* for overall breast cancer, *RNF175* and *NCKAP1L* for ER-positive disease, and *PHAX, SMARCA2, NTRK3, EML5*, and *MED23* for ER-negative disease. Because the observed effects did not reach the threshold significance level following the adjustment of probability for multiple testing, we cannot exclude the possibility that the excess of missense variants, in the absence of enrichment for LoF variants, may be a false-positive result. However, these findings may also suggest that some of these genes are intolerant of LoF variants. When looking at the best candidate genes for overall breast cancer, pLI (probability of being loss-of-function intolerant) scores and observed/expected (o/e) scores available on gnomAD seem to suggest that this may be the case for *SIPAL1*, with pLI = 1, o/e = 0.09 (0.05–0.17). Interestingly, we did not observe any LoF variants in this gene in any of our analyses. On the other hand, *ERCC2* (pLI = 0, o/e = 0.82 (0.62–1.09)) and *TMEM161A* (pLI = 0; o/e = 0.53 (0.35–0.82)) scores seem to indicate that these genes are not LoF intolerant. For ER-subtype-specific associations, LoF intolerant scores were observed for *NCKAP1L* (pL1 = 0.85, o/e = 0.2 (0.13–0.33))*, SMARCA2* (pL1 = 1, o/e = 0.12 (0.08–0.2)), and *NTRK3* (pL1 = 1, o/e = 0.12 (0.06–0.26)) (https://gnomad.broadinstitute.org, accessed on 9 February 2022). Among these genes, *ERCC2* is of particular interest, as it plays an important role in the nucleotide excision repair pathway, and common polymorphisms in this gene have been associated with risk of various types of cancers [40]. The evidence supporting *ERCC2* as a more penetrant breast cancer susceptibility gene, however, is not convincing [41].

Our study also included the analysis of 32 well-established or suspected susceptibility genes included in breast cancer gene panels. For many genes in such panels, the evidence of an association with cancer is often weak, and, therefore, further validation is required to confirm their contribution to breast cancer risk [4]. The current validation study successfully detected confirmed breast cancer susceptibility genes such as *ATM*, *CHEK2*, and *PALB2*. The ORs calculated in our study of these frequently mutated genes are similar to those calculated in recent publications [8,18,42,43,44,45] according to analyses of LoF and rare missense variants. Similar to the current analysis, the study by Dorling et al. [18] did not detect a statistically significant association of missense variants in *PALB2* (OR = 0.96 (95% CI 0.87–1.06)) with breast cancer risk. It should be noted, however, that there is a significant overlap of the current study with the Dorling et al. dataset, which may contribute towards explaining these observations. For the remaining suspected susceptibility genes, including genes confirmed to be associated with breast cancer risk or predisposition to specific cancer syndromes, as studied by Lee et al. [46], Dorling et al. [18], and/or Hu et al. [27] (*BARD1*, *CDH1*, *PTEN*, *RAD51C*, *RAD51D*, *STK11*, *TP53*), the frequencies of LoF and rare missense variants were too low in our study to detect or support an association with breast cancer risk. 

### Limitations and Weaknesses

It should be noted that this study has some limitations. Firstly, the discovery step involved WES of samples from breast cancer cases followed by analysis of the resulting data with available WES or WGS control data. This study design was motivated by the possibility of taking advantage of several available datasets. However, many factors resulting from this choice may have affected our ability to identify likely candidate genes at the discovery step, such as: (1) The matched control data were derived from different exome enrichment methods. These methods do not have the same efficiency in terms of probe hybridization and exon capture. This may have contributed to uneven exon sequencing depth, which in turn could have affected the identification of variants [47]; (2) For the GoNL control dataset, exome data were extracted from WGS data. Although it is recognized that WGS outperforms WES, and provides a higher proportion of covered transcripts at a lower sequencing coverage and no strand bias [48], the 13× sequencing coverage for these samples may not have been sufficient for the purpose of identifying rare variants. In an attempt to minimize these impacts, several calling scenarios for the joint genotyping of case and control samples, taking into consideration ethnicity and capture technology, were performed in order to ensure that the most comprehensive variety of calling scenarios were covered. Five different variant filtering and prioritization strategies were also performed in an attempt to build the best candidate gene list for the validation step. Despite this, it cannot be excluded that insufficient or non-uniform sequencing coverage across exons between case and controls [49], batch effects, and population or cryptic stratification [50] could have had an impact on variant calling, which may not have been resolved at the bioinformatics level [48]. We also acknowledge the limitations of the currently available in silico tools for the prediction of pathogenic missense variants, and how these limitations could have impacted our gene selection.

Secondly, while the replication stage was quite large, important susceptibility may not have been selected at the discovery stage due to the limitations mentioned above. Moreover, although the validation stage was of a large enough scale to identify genes with effect sizes similar to those of *CHEK2*, *ATM*, and *PALB2*, the power was too low to detect genes with the characteristics of *RAD51C*, *RAD51D*, or *BARD1.* Indeed, the associations with LoF variants in these genes were all non-significant, despite the fact that the estimated ORs (~2) were comparable with those previously reported. Thus, if a susceptibility gene with the characteristics of the latter was among those identified at the discovery stage, the validation step would not have had sufficient power to confirm it. We therefore conclude that it is unlikely that any genes in the replication set with combined LoF frequencies comparable with *ATM* or *CHEK2* are associated with moderate risk, but that moderate-risk genes with lower LoF frequencies could be present.

Finally, our study was based on WES, which does not consider non-coding regulatory regions or large genomic rearrangements. Regulatory variants have been shown to contribute to the heritability of several diseases. In particular, the common susceptibility variants for breast cancer identified through GWAS are mostly located in regulatory regions [51].

## 5. Conclusions

We found nominal evidence of association with breast cancer overall for LoF variants in three genes (*ZFAND1*, *TMEM206/PACC1*, and *TYRO3),* with ER-negative breast cancer in two genes, *DNAH11* and *PARP2*, and with ER-positive disease in four genes *LAMC3*, *MTMR11, EPN3*, and *SLC22A10*. Analysis of rare missense variants identified evidence of associations of *TMEM161A*, *SIPA1L1*, and *ERCC2* with overall breast cancer, *RNF175* and *NCKAP1L* with ER-positive disease, and *SLC22A10*, *PHAX*, *SMARCA2*, *EML5*, *NTRK3*, and *MED23* with ER-negative disease. Our study design shows that we had the power to detect genes with effect sizes similar to some confirmed breast cancer susceptibility genes, such as *ATM* and *CHEK2*. The fact that we did not identify similar novel genes suggests that the remaining breast cancer susceptibility could be explained by lower frequency variants. The outcome of our study thus provides crucial information with regards to the planning of future sequencing efforts.

In order to efficiently identify novel lower frequency variants, larger collaborative sequencing efforts will be needed. The data generated in the current project will most certainly be useful for combining with other similar large-scale breast cancer studies in order to obtain better power to detect such variants in moderate-risk genes.

## Figures and Tables

**Figure 1 cancers-14-03363-f001:**
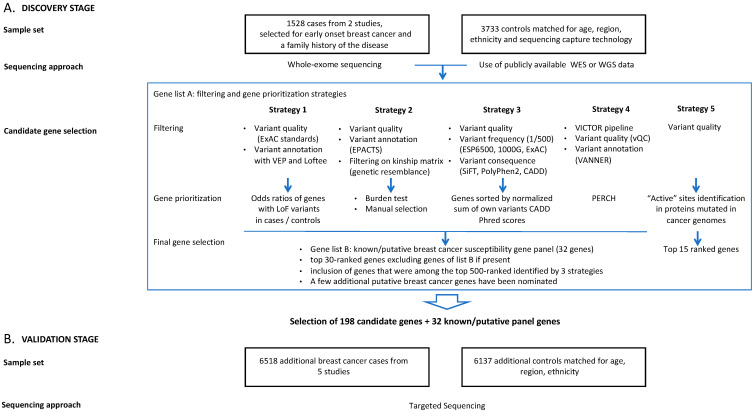
Study design for breast cancer susceptibility gene discovery by whole-exome sequencing (**A**) and validation by targeted enrichment sequencing (**B**).

**Figure 2 cancers-14-03363-f002:**
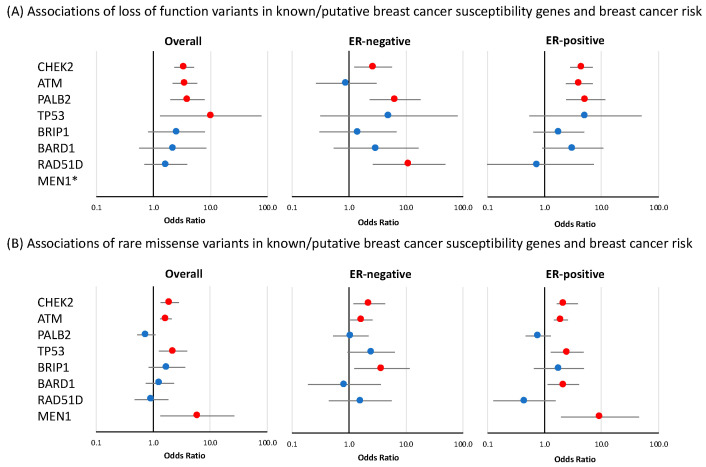
Risk of breast cancer overall and tumor subtypes associated with loss-of-function and missense variants in known or suspected breast cancer susceptibility genes included in breast cancer gene panels. Shown are odds ratios (ORs) and 95% confidence intervals (CIs) for breast cancer overall, estrogen receptor (ER)-negative breast cancer, and ER-positive breast cancer associated with loss-of-function variants (**A**) and missense variants (**B**) in genes for which evidence of association was observed in at least one of the analyses. Red circles indicate *p*-value < 0.05. * no loss-of-function variants were observed in *MEN1*.

**Figure 3 cancers-14-03363-f003:**
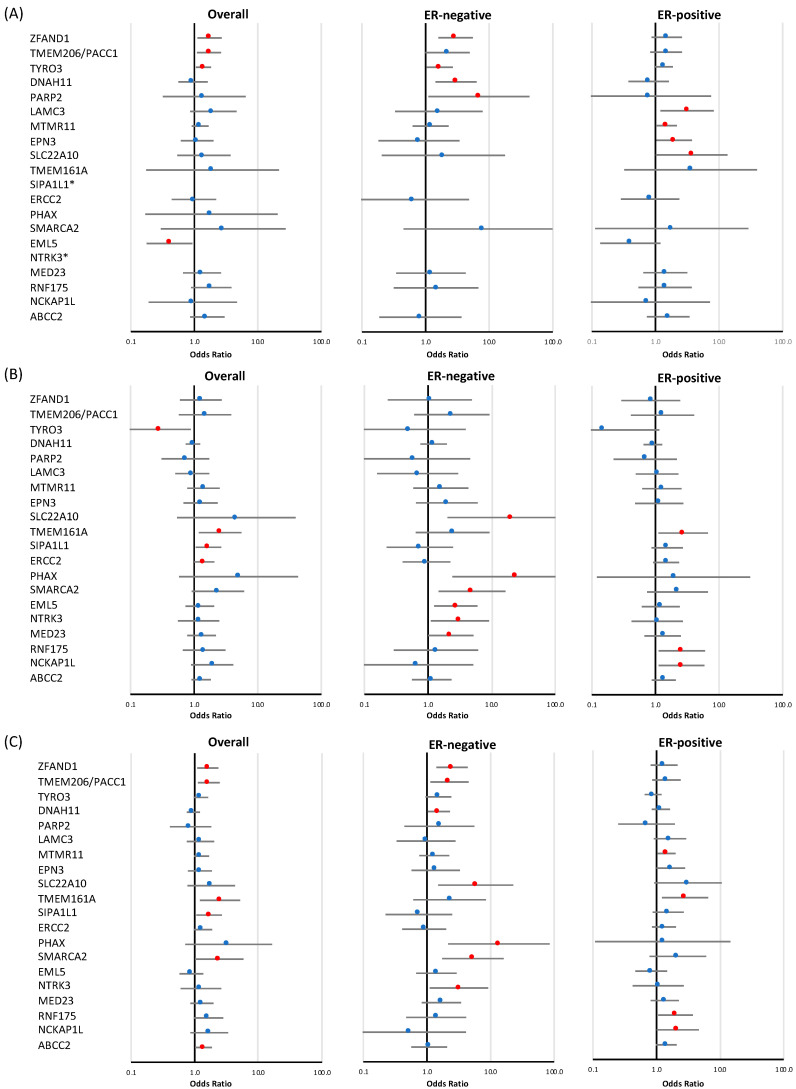
Risk of breast cancer overall and tumor subtypes associated with loss-of-function and missense variants in a subset of targeted genes identified at the discovery stage. Shown are odds ratios (ORs) and 95% confidence intervals (CIs) for breast cancer overall, estrogen receptor (ER)-negative breast cancer, and ER-positive breast cancer associated with loss-of-function variants (**A**), missense variants (**B**), and combined loss-of-function and missense variants (**C**) in genes for which evidence of association was observed in at least one of the analyses. Red circles indicate *p*-value < 0.05. * no loss-of-function variants were observed in *SIPA1L1* and *NTRK3*.

**Table 1 cancers-14-03363-t001:** Associations of loss-of-function variants in 32 genes included in breast cancer gene panels with overall breast cancer risk and ER-negative and ER-positive breast cancer risk. Shown are odds ratios (ORs), 95% confidence intervals (CIs), and *p*-values. Associations with *p* < 0.05 are denoted in bold.

		Overall					ER-Negative					ER-Positive				
Gene	Number of Carriers	Number of Carriers	OR	95% CI		*p*-Value	Number of Carriers	OR	95% CI		*p*-Value	Number of Carriers	OR	95% CI		*p*-Value
	Controls (n = 6019)	Cases (n = 6211)					Cases (n = 808)					Cases (n = 2764)				
AKT1	1	1	0.89	0.06	14.25	0.935	0	0.00	0.00	Inf	0.997	1	1.74	0.11	27.90	0.694
ATM	**20**	**74**	**3.55**	**2.16**	**5.82**	**5.44 × 10^−7^**	3	0.89	0.26	3.02	0.851	**42**	**4.06**	**2.37**	**6.97**	**3.39 × 10^−7^**
BARD1	4	11	2.54	0.81	7.98	0.110	2	2.98	0.54	16.57	0.212	7	3.13	0.91	10.71	0.069
BRE/BABAM2	0	0	-	-	-	-	0	-	-	-	-	0	-	-	-	-
BRIP1	8	14	1.64	0.69	3.93	0.262	2	1.44	0.30	6.84	0.649	7	1.78	0.64	4.97	0.274
CDH1	0	4	0.00	0.00	Inf	0.906	2	0.00	0.00	Inf	0.994	1	0.00	0.00	Inf	0.994
CHEK2	**32**	**111**	**3.47**	**2.33**	**5.15**	**7.62 × 10^−10^**	**9**	**2.65**	**1.24**	**5.70**	**0.012**	**56**	**4.44**	**2.82**	**6.99**	**1.23 × 10^−10^**
EPCAM	3	1	0.33	0.04	3.22	0.343	0		0.00	Inf	0.995	1	0.60	0.06	5.77	0.658
FAM175A/ABRAXAS1	0	0	-	-	-	-	0	-	-	-	-	0	-	-	-	-
FANCC	15	13	0.86	0.41	1.81	0.692	1	0.63	0.08	4.91	0.662	6	1.00	0.37	2.66	0.995
FANCM	39	42	1.03	0.67	1.60	0.894	8	1.52	0.70	3.30	0.295	20	1.09	0.63	1.90	0.749
GEN1	2	3	1.54	0.26	9.26	0.636	1	7.68	0.45	131.19	0.159	2	3.56	0.32	39.34	0.300
MEN1	0	0	-	-	-	-	0	-	-	-	-	0	-	-	-	-
MLH1	1	1	0.89	0.06	14.25	0.935	0		0.00	Inf	0.997	1	1.74	0.11	27.90	0.694
MRE11A/MRE11	0	0	-	-	-	-	0	-	-	-	-	0	-	-	-	-
MSH2	0	2	0.00	0.00	Inf	0.934	0	-	-	-	-	1	0.00	0.00	Inf	0.994
MSH6	**47**	**19**	**0.36**	**0.21**	**0.62**	**2.07 × 10^−4^**	5	0.55	0.22	1.39	0.204	**11**	**0.42**	**0.22**	**0.81**	**9.00 × 10^−3^**
MUTYH	5	4	0.78	0.21	2.93	0.718	1	2.06	0.21	20.54	0.537	1	0.73	0.07	7.19	0.784
NBN	27	19	0.65	0.36	1.17	0.150	3	0.61	0.18	2.02	0.416	13	0.90	0.46	1.76	0.766
NF1	18	19	0.95	0.50	1.81	0.867	3	0.86	0.25	2.92	0.805	13	1.26	0.62	2.59	0.522
PALB2	**10**	**42**	**3.95**	**1.98**	**7.88**	**9.68 × 10^−5^**	**7**	**6.45**	**2.31**	**17.98**	**3.72 × 10^−4^**	**22**	**5.27**	**2.39**	**11.61**	**3.83 × 10^−5^**
PIK3CA	0	0	-	-	-	-	0	-	-	-	-	0	-	-	-	-
PMS2	27	16	0.53	0.29	0.98	0.044	3	0.57	0.17	1.87	0.351	8	0.53	0.24	1.16	0.111
PTEN	1	2	1.78	0.16	19.65	0.637	0	0.00	0.00	Inf	0.997	1	1.74	0.11	27.90	0.694
RAD50	9	9	0.97	0.38	2.44	0.943	1	1.05	0.13	8.59	0.967	8	2.45	0.90	6.71	0.081
RAD51C	3	6	1.89	0.47	7.55	0.370	2	3.90	0.64	23.88	0.141	2	1.19	0.20	7.14	0.847
RAD51D	3	7	2.18	0.56	8.45	0.258	**5**	**11.26**	**2.60**	**48.68**	**1.19 × 10^−3^**	1	0.74	0.08	7.31	0.795
RECQL	9	12	1.28	0.54	3.05	0.576	0	0.00	0.00	Inf	0.990	6	1.64	0.55	4.92	0.379
RINT1	6	5	0.76	0.23	2.50	0.655	0	0.00	0.00	Inf	0.992	2	0.59	0.12	2.90	0.512
STK11	0	0	-	-	-	-	0	-	-	-	-	0	-	-	-	-
TP53	**1**	**11**	**10.16**	**1.31**	**78.67**	**0.026**	1	5.00	0.31	80.03	0.255	3	5.24	0.54	50.39	0.152
XRCC2	0	0	-	-	-	-	0	-	-	-	-	0	-	-	-	-

**Table 2 cancers-14-03363-t002:** Associations with breast cancer risk of loss-of-function and rare missense variants in top candidate genes identified at the discovery step. Shown are odds ratios (ORs), 95% confidence intervals (CIs), and *p*-values for breast cancer overall, estrogen receptor (ER)-negative breast cancer, and ER-positive breast cancer. Associations with *p* < 0.05 are denoted in bold.

		Overall					ER-Negative					ER-Positive				
	Number of Carriers		OR	95% CI		*p*-Value	Number of Carriers	OR	95% CI		*p*-Value	Number of Carriers	OR	95% CI		*p*-Value
	Controls n = 6019	Cases n = 6211					Cases n = 808					Cases n = 2764				
**(A) Loss-of-function variants**																
ZFAND1	**31**	**59**	**1.73**	**1.12**	**2.68**	**0.014**	**16**	**2.96**	**1.59**	**5.50**	**6.37 × 10^−4^**	24	1.50	0.87	2.60	0.146
TMEM206/PACC1	**33**	**56**	**1.70**	**1.11**	**2.62**	**0.016**	8	2.21	0.99	4.93	0.052	21	1.47	0.83	2.61	0.186
TYRO3	**89**	**131**	**1.40**	**1.06**	**1.83**	**0.016**	**22**	**1.66**	**1.03**	**2.68**	**0.038**	58	1.34	0.95	1.88	0.092
DNAH11	28	27	0.96	0.56	1.62	0.865	**10**	**3.00**	**1.42**	**6.32**	**3.88 × 10^−3^**	10	0.78	0.37	1.63	0.507
PARP2	3	4	1.43	0.32	6.40	0.641	**2**	**6.89**	**1.11**	**42.78**	**0.038**	1	0.75	0.08	7.45	0.808
LAMC3	8	17	2.00	0.86	4.64	0.107	2	1.61	0.33	7.87	0.553	**11**	**3.14**	**1.20**	**8.20**	**0.020**
MTMR11	75	95	1.24	0.91	1.68	0.174	11	1.20	0.63	2.29	0.583	**49**	**1.49**	**1.03**	**2.17**	**0.037**
EPN3	21	24	1.10	0.61	1.99	0.742	2	0.79	0.18	3.40	0.747	**18**	**1.97**	**1.03**	**3.76**	**0.039**
SLC22A10	7	10	1.41	0.54	3.72	0.484	1	1.91	0.21	17.73	0.571	**6**	**3.70**	**1.01**	**13.57**	**0.048**
TMEM161A	1	2	1.94	0.18	21.43	0.590	0	0.00	0.00	Inf	0.997	2	3.56	0.32	39.34	0.300
SIPA1L1	0	0	-	-	-	-	0	-	-	-	-	0	-	-	-	-
ERCC2	12	12	0.99	0.44	2.20	0.972	1	0.62	0.08	4.83	0.646	5	0.83	0.29	2.38	0.724
PHAX	1	2	1.86	0.17	20.52	0.612	0	0.00	0.00	Inf	0.997	0		0.00	Inf	0.995
SMARCA2	1	3	2.85	0.30	27.42	0.365	1	7.68	0.45	131.19	0.159	1	1.80	0.11	28.81	0.678
EML5	18	8	0.41	0.18	0.94	0.035	0	0.00	0.00	Inf	0.986	4	0.40	0.14	1.19	0.100
NTRK3	0	1	0.00	0.00	inf	0.930	0	-	-	-	-	0	-	-	-	-
MED23	14	20	1.32	0.67	2.62	0.424	3	1.21	0.35	4.27	0.763	11	1.43	0.65	3.17	0.375
RNF175	11	21	1.85	0.89	3.85	0.099	2	1.46	0.32	6.75	0.628	7	1.42	0.54	3.72	0.478
NCKAP1L	3	3	0.95	0.19	4.71	0.948	0	0.00	0.00	Inf	0.995	1	0.73	0.07	7.19	0.784
ABCC2	16	26	1.60	0.86	2.99	0.140	2	0.83	0.19	3.67	0.804	12	1.59	0.74	3.42	0.240
**(B) Rare missense variants**																
ZFAND1	12	16	1.27	0.60	2.70	0.527	2	1.07	0.23	4.89	0.930	5	0.85	0.29	2.45	0.759
TMEM206/PACC1	7	11	1.48	0.57	3.83	0.416	3	2.37	0.61	9.27	0.214	5	1.29	0.41	4.06	0.669
TYRO3	13	4	0.29	0.10	0.89	0.031	1	0.51	0.07	3.93	0.516	1	0.15	0.02	1.15	0.068
DNAH11	116	115	0.95	0.74	1.24	0.726	21	1.22	0.76	1.97	0.406	52	0.91	0.65	1.28	0.603
PARP2	12	9	0.73	0.31	1.73	0.470	1	0.59	0.08	4.61	0.614	4	0.69	0.22	2.17	0.524
LAMC3	21	20	0.93	0.51	1.73	0.826	2	0.69	0.16	3.00	0.620	10	1.06	0.49	2.30	0.888
MTMR11	19	28	1.40	0.78	2.52	0.256	5	1.59	0.59	4.31	0.362	13	1.26	0.62	2.56	0.524
EPN3	18	23	1.26	0.68	2.35	0.460	4	1.98	0.65	6.05	0.233	8	1.15	0.48	2.74	0.751
SLC22A10	1	5	4.61	0.54	39.46	0.163	**3**	**20.76**	**2.03**	**211.95**	**0.011**	0	0.00	0.00	Inf	0.995
TMEM161A	**9**	**24**	**2.56**	**1.19**	**5.51**	**0.016**	3	2.44	0.64	9.26	0.190	**11**	**2.73**	**1.11**	**6.71**	**0.029**
SIPA1L1	**29**	**50**	**1.68**	**1.06**	**2.67**	**0.026**	3	0.74	0.22	2.48	0.631	22	1.53	0.87	2.69	0.140
ERCC2	**49**	**73**	**1.45**	**1.01**	**2.08**	**0.047**	6	0.95	0.40	2.25	0.906	31	1.47	0.92	2.35	0.108
PHAX	1	5	4.99	0.58	42.78	0.143	**3**	**24.27**	**2.42**	**243.37**	**6.69 × 10^−3^**	1	1.92	0.12	30.68	0.646
SMARCA2	6	15	2.36	0.92	6.09	0.076	**5**	**4.92**	**1.47**	**16.48**	**9.69 × 10^−3^**	7	2.23	0.74	6.71	0.153
EML5	26	33	1.23	0.73	2.06	0.432	**9**	**2.74**	**1.25**	**6.01**	**0.012**	13	1.21	0.61	2.41	0.587
NTRK3	13	15	1.18	0.56	2.48	0.669	**5**	**3.17**	**1.11**	**9.11**	**0.032**	7	1.06	0.42	2.69	0.897
MED23	25	34	1.31	0.78	2.20	0.305	**8**	**2.29**	**1.01**	**5.18**	**0.048**	15	1.30	0.67	2.51	0.434
RNF175	11	16	1.43	0.66	3.09	0.361	2	1.34	0.29	6.14	0.710	**12**	**2.59**	**1.12**	**6.00**	**0.026**
NCKAP1L	10	20	1.92	0.90	4.10	0.093	1	0.65	0.08	5.16	0.686	**13**	**2.57**	**1.12**	**5.91**	**0.027**
ABCC2	57	76	1.29	0.91	1.82	0.150	9	1.14	0.56	2.34	0.714	36	1.36	0.88	2.09	0.163
**(C) Combined analysis: loss-of-function and rare missense variants**																
ZFAND1	**43**	**75**	**1.61**	**1.1**	**2.34**	**0.014**	**18**	**2.47**	**1.40**	**4.36**	**1.78 × 10^−3^**	29	1.32	0.82	2.15	0.257
TMEM206/PACC1	**40**	**67**	**1.67**	**1.12**	**2.47**	**0.011**	**11**	**2.26**	**1.13**	**4.51**	**0.021**	26	1.43	0.86	2.39	0.169
TYRO3	102	135	1.25	0.97	1.63	0.090	23	1.51	0.95	2.41	0.082	59	1.18	0.85	1.64	0.324
DNAH11	144	142	0.95	0.75	1.21	0.696	**31**	**1.53**	**1.03**	**2.29**	**0.037**	62	0.89	0.65	1.21	0.449
PARP2	15	13	0.86	0.41	1.81	0.694	3	1.57	0.44	5.58	0.484	5	0.70	0.25	1.96	0.497
LAMC3	29	37	1.23	0.76	2.01	0.402	4	0.96	0.33	2.80	0.947	21	1.63	0.91	2.93	0.104
MTMR11	94	123	1.27	0.97	1.67	0.082	16	1.30	0.75	2.24	0.345	**62**	**1.44**	**1.03**	**2.01**	**0.032**
EPN3	38	47	1.21	0.79	1.86	0.385	6	1.37	0.57	3.30	0.484	26	1.67	1.00	2.81	0.051
SLC22A10	8	15	1.82	0.77	4.31	0.171	4	5.84	1.51	22.69	0.011	6	3.12	0.92	10.61	0.068
TMEM161A	**10**	**26**	**2.50**	**1.2**	**5.19**	**0.014**	3	2.26	0.61	8.44	0.224	**13**	**2.83**	**1.22**	**6.55**	**0.015**
SIPA1L1	**29**	**50**	**1.68**	**1.06**	**2.67**	**0.026**	3	0.74	0.22	2.48	0.631	22	1.53	0.87	2.69	0.140
ERCC2	60	83	1.34	0.96	1.88	0.083	7	0.90	0.41	2.01	0.802	35	1.32	0.86	2.04	0.210
PHAX	2	7	3.41	0.71	16.4	0.126	**3**	**13.49**	**2.15**	**84.71**	**5.50 × 10^−3^**	1	1.26	0.11	14.53	0.854
SMARCA2	**7**	**18**	**2.43**	**1.01**	**5.83**	**0.046**	**6**	**5.27**	**1.73**	**16.05**	**3.40 × 10^−3^**	8	2.17	0.78	6.03	0.138
EML5	44	41	0.89	0.58	1.36	0.581	9	1.40	0.67	2.91	0.366	17	0.83	0.47	1.47	0.519
NTRK3	13	16	1.25	0.6	2.6	0.552	**5**	**3.17**	**1.11**	**9.11**	**0.032**	7	1.06	0.42	2.69	0.897
MED23	39	53	1.29	0.85	1.96	0.225	10	1.69	0.83	3.43	0.148	26	1.35	0.82	2.25	0.242
RNF175	22	37	1.64	0.97	2.79	0.066	4	1.40	0.47	4.12	0.545	**19**	**2.00**	**1.07**	**3.75**	**0.031**
NCKAP1L	13	23	1.69	0.86	3.35	0.130	1	0.53	0.07	4.09	0.542	**14**	**2.16**	**1.01**	**4.65**	**0.048**
ABCC2	**72**	**102**	**1.38**	**1.02**	**1.87**	**0.039**	11	1.08	0.56	2.06	0.821	48	1.42	0.98	2.07	0.067

## Data Availability

The complete dataset will not be made publicly available due to restraints imposed by the ethics committees of individual studies; requests for data can be made to the corresponding author or to the Data Access Coordination Committee (DACCs) of the Breast Cancer Association Consortium (https://bcac.ccge.medschl.cam.ac.uk/, accessed on 1 June 2022).

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
