# Peer review of "Uncovering the Contribution of Moderate-Penetrance Susceptibility Genes to Breast Cancer by Whole-Exome Sequencing and Targeted Enrichment Sequencing of Candidate Genes in Women of European Ancestry"

_cancers, 2022, doi:10.3390/cancers14143363_

Round 1
Reviewer 1 Report
This manuscript addressed a timely topics, the identification of genes involved in breast cancer (BC) development. The authors examined a large number of BC patients and control people and proposed previously unappreciated genes potentially involved in BC development. The reviewer would like to recommend the publication of this manuscript as it is
Author Response
We thank the reviewer for their kind comment.
Reviewer 2 Report
Comment 1.
Even though the odds ratio per gene in association with breast cancer risk is important, it is often more important to know the odds ratio per variant. Even though the odds ratio per variant is not statistically significant due to sizes of case and control population, knowing the odds ratio per variant (even when there are insignificant p-values) is very important, regardless of the average odds ratio per gene. Hence, the authors should provide the supplementary table showing all of the odds ratio per variant, and include a subsection explaining the variants with high odds ratio in the revised manuscript.
Author Response
Although we appreciate the reviewer’s comment with regard to the relevance of individual variant associations, our study design was not intended to examine individual variants because the vast majority of these variants are unique and/or too rare to calculate an odds ratio. Therefore, we have chosen a gene burden test in order to improve the power to identify associations. However, in an attempt to address the reviewer's comment, we performed an analysis of individual variants for those observed in at least 3 carriers, at least two of which were observed in breast cancer cases, and for which providing an odds ratio was more relevant. Our analysis therefore focused on 150 LoFs among the 1,696 observed, and 373 rare missense variants among the 5,741 observed. These data are presented in a new supplementary table S11. We have also added the following paragraph in the results section to incorporate and justify this individual variant analysis approach:
3.2.4 Individual LoF and missense variant analysis
As mentioned in the Materials and Methods section, the main association analysis performed in our study was a gene burden analysis. This approach increases the power to detect an association and is used by a vast majority of association studies, including those recently published by Dorling et al [18] and Hu et al. [27]. Nevertheless, it has been shown that single variants can show evidence of association when a weaker signal is obtained at the gene level [28,29]. We therefore examined whether certain individual LoF or missense variants with higher frequencies showed evidence of association and whether they were driving the observed associations for a given gene. In this analysis, we only examined individual variants observed in at least 3 carriers, at least two of which were observed in breast cancer cases. As shown in supplementary Table S11A, a total of seven LoF variants in six different genes showed evidence of association with overall breast cancer (nominal p-value<0.05). Three of these variants were observed in known breast cancer susceptibility genes, one in ATM (c.1564_1565delGA, OR=8.77 [CI 95% 1.11-69.29], p-value=0.040) that is reported as pathogenic in the ClinVar database [30] and two in CHEK2 (c.1100delC, OR=3.08 [CI 95% 2.01-4.3], p-value=2.83x10-7; c.444+1G>A, OR=8.12 [CI 95% 1.03-64.13], p-value=0.047). Three other LoF variants showing evidence of association in our single variant analysis were located in the top 3 candidate genes identified through the gene burden test (ZFAND1 c.139-3_139delCAGG, OR=1.76 [CI 95% 1.11-2.81], p-value=0.017), TMEM206/PACC1 c.631delC OR=1.81 [CI 95% 1.15-2.86], p-value=0.011 and TYRO3 c.308_308+1insCCTGAAGTCA OR=1.55 [CI 95% 1.15-2.09], p-value=3.93x10-3). These 3 variants have a much higher frequency than other such variants identified in these genes. They clearly contribute to the association observed at the gene level. Lastly, one Lof variant in the TRPM4 gene had a very high OR, while no evidence of association was observed for this gene when all LoF variants were analysed in aggregate (TRPM4, c.2254C>T, OR=10.64 [CI 95% 1.37-82.46], p-value=0.024). A similar analysis performed on rare missense variants (Table S11B) identified only 3 variants with nominally significant p-values: one in ATM (p.L2307F, OR=5.94 [CI 95% 2.06-17.16], p-value=9.85x10-4); one in PMS2 (p.I18T, OR=4.74 [CI 95% 1.02-22.01], p-value=0.047); and one in RIC1 (p.R635H, OR=6.07 [CI 95% 1.36-27.17], p-value=0.018). With the exception of the ATM gene, PMS2 and RIC1 showed no evidence of association in the gene burden test. Even though these variants showed a deleteriousness score according to the BayesDel algorithm, caution should be taken when interpreting such missense variants because their impact may differ greatly depending on their position in the gene (e.g. conserved or functional domains).
Along with this new text, three additional references have been added to the mansucript:
[28] Lee, S.; Abecasis, G.R.; Boehnke, M.; Lin, X. Rare-variant association analysis: study designs and statistical tests. Am J Hum Genet 2014, 95, 5-23. doi:10.1016/j.ajhg.2014.06.009.
[29] Liu, D.J.; Peloso, G.M.; Zhan, X.; Holmen, O.L.; Zawistowski, M.; Shuang, F.; Nikpay, M.; Auer, P.L.; Goel, A; Zhang, H.; et al. Meta-analysis of gene-level tests for rare variant association. Nat Genet 2014, 46, 200-204. doi: 10.1038/ng.2852.
[30] Landrum, M.J.; Lee, J.M.; Benson, M.; Brown, G.R.; Chao, C.; Chitipiralla, S.; Gu, B.; Hart, J.; Hoffman, D.; Jang, W.; et al. ClinVar: improving access to variant interpretations and supporting evidence. Nucleic Acids Res 2018, 46, D1062-D1067. doi: 10.1093/nar/gkx1153.
Reviewer 3 Report
I enjoyed reading this manuscript. It is of high quality. Not so sure about divide highly heterogeneous BrCa into ER positive and ER negative camps. PAM50-based molecular grouping might be a better choice for studying gene signatures in different groups.
I have gone through the gene lists identified by this study. Many of them make sense like the well-known BRCA1/2, ATM and mediator genes. However, some well known genes are missing, such as p27.
I like the part the author discussed the limitation of this study. Besides the bias of the grouping method aforementioned, the bias from asynchronized cell cycle from tumor sample may obscure the differential expression of cell cycle -related genes.
One more thing, we have been using PAM-50 genes to classify BrCa types. Have some of these 50 genes shown up in some of your lists? This WES/WGS study may help to reveal the function of some genes in PAM-50.
Author Response
R: We thank the reviewer for their kind comments. We acknowledge that ER positive and negative tumors are heterogenous but unfortunately data on HER2 (a significant number of these breast cancers were diagnosed before the collection of this data was standard practice) and grade have not been systematically collected and the incompleteness of these data does not allow us to perform a thorough analysis.
We have now acknowledged this in the discussion by adding the following sentence: In addition to clinical estrogen receptor subtyping, analyses performed on the molecular subtypes of breast cancer, namely: Luminal A, Luminal B, Her2-enriched, Basal-like, and Normal-like, would have been relevant in the context of this analysis given the recognized heterogeneity of ER-positive and ER-negative tumors. However, the limited number of samples for which we had data did not allow us to perform a meaningful analysis for these molecular subtypes.
As noted by the reviewer, p27 (CDKN1B) was absent from the list of genes that were selected for validation. This is still the case even when we broaden our selection to the top 500 genes. Although p27 is a well-known gene involved in many types of cancer, most of the literature has reported p27 mutations in tumor samples rather than germline variants as those described in the current study. This could possibly explain why this particular gene (and others known to be somatically mutated in tumors) has not been identified through our study design.
To answer the reviewer’s question about whether any PAM50 genes have showed up in our lists. Only three PAM50 genes (CEP55, EGFR, MAPT) were selected for validation following the WES stage, however our analysis did not reveal any associations (p<0.05) for these genes. No other PAM50 genes showed up in our final list of candidate genes. Indeed, the question whether the genes that are mutated in the germline are also those that are most important for the classification of subtypes and subsequent prognosis is an intriguing question, to which we could only speculate based on our study.
Reviewer 4 Report
In this paper the use of technology advances in next generation sequencing offer access and solutions in identifying genes with susceptibility variants in women with breast cancer of European ancestry. In terms of computational power to perform analysis the collaboration between five institutions offers scalability, security and performance. We need to acknowledge that the authors strive to close the gap between the disconnection of high-throughput sequencing (HTS) data and the analysis required to facilitate biological understanding. A large scale project such as this can only be achieved through the use of appropriate study design, robust statistical methods and validation. With all its ‘limitations and weaknesses’, as mentioned by the authors, the essential takeaway of this paper is to apply the lessons learned and come up with new solutions and interpretation methods. Because with every stepping stone put in place we only get closer in solving the bottleneck in biomarker discovery research that will help identify woman with a high breast cancer risk.
Author Response

(The authors gave the same response as above.)
